# Distinct Computations Emerge From Compositional Curricula In-Context

## Abstract

In-context learning (ICL) research often considers learning a single class of function in-context through a uniform sample of input-output pairs. However, natural language data often has more complex structural correlations, such as the composition of information in a given context. Here, we study such compositional structure in context with a toy modular-arithmetic task and investigate how the in-context curriculum of constituent function examples may alter the computations a transformer learns to solve compositional tasks. We compare models trained with varying in-context curricula of subtasks and the composite task examples. We show that models trained with subtasks in-context generalize to unseen compositional tasks by building an inner representation of the intermediate computation of subtasks. Finally, we find that the model often exhibits a continuous spectrum of a compositional strategy, rather than discrete modes, which are modulated by curriculum design.

## 1 Introduction

Many complex real-world concepts and tasks are constructed by composing multiple constituents. This notion of systematic compositionality has been extensively studied (Chomsky, 1999; Frege, 1948; Szabó, 2024) and is argued to be a key feature of flexible intelligence, enabling "infinite use from finite means". A natural language corpus encompasses a variety of structures that reflect such compositional nature: introducing simpler elements in the beginning that build to yield more complex interactions later. For example, many essays and informative writings contain paragraphs, each of which introduces supporting arguments, and at the end follows a conclusion paragraph that composes the previous points. Instructions or math problem solutions similarly contain compositional structure. Concurrently, previous works highlight the importance of distributional properties for the emergence of ICL, pointing out that data characteristics play a crucial role in determining the computations the model develops Chan et al. (2022). With this perspective, one might hypothesize that the properties of the structure in which the data is present in context might influence the model computation. Inspired by these, we study one instance of compositional structure in-context, namely subtask curricula —in-context examples of constituent functions and their composition— and demonstrate that it can give rise to distinct computations that the model learns. Concretely,

- We design an algorithmic task based on a modular double exponential to study the generalization behavior of transformer models trained with varying in-context compositional curricula.

- We show evidence that the model utilizes subtask information from the curriculum to solve a compositional task using linear probes and a causal behavioral experiment.

- We demonstrate that the different degrees of in-context correlations between subtasks and the compositional task modulate a diverse mixture of compositional strategy and standard few-shot learning strategy in context.

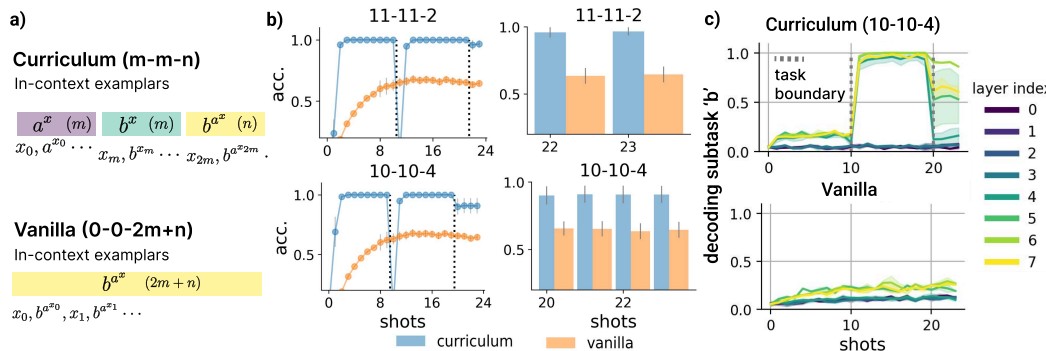

Figure 1: **Overview of the setup and key results. a)** Task schema. In curriculum training, each training sequence is composed of $m$ exemplars for two single-exponential tasks defined by $a$ and $b$, followed by $n$ composite double-exponential task exemplars. In vanilla training, the model is trained with a sequence of $2m + n$ in-context exemplars for the double-exponential task defined by task parameters $(a, b)$. **b)** Comparison of the accuracy for curriculum training and vanilla training in the challenging test cases of $a = 30$ (top: $m = 11, n = 2$, bottom: $m = 10, n = 4$). Right: Throughout the entire context length. Left: Zooming into the last compositional block $n$ and corresponding context shots for the vanilla model. **c)** In-context curricula promote representation of intermediate task parameters. We show linear probe decoding accuracy of intermediate computation values from unseen evaluation sequences. The curriculum-trained model represents the task parameter in the compositional task block (last 4 shots) while the vanilla-trained model does not.

## 2 EXPERIMENTAL SETUP

### 2.1 TASK

**Modular Double Exponential Task.**    Modular arithmetic tasks have been extensively used in prior works (He et al., 2024; Nanda et al., 2023a; Power et al., 2022; Zhong et al., 2023) to understand the inner working of transformers, as it offers deterministic functional mapping and a constrained vocabulary size, which allows controlled study. We use a composition of two exponential functions, namely $b^{a^x} \bmod P$, which we refer to as the modular double exponential task. Its greater complexity compared to linear operations helps us to distinguish failure modes of the compositional generalization, while the modular arithmetic retains the controllability of our experiments. We further elaborate on characteristics of the modular double exponential task and why we chose this task in Appendix A.

**Task Sampling.**    For every example sequence, we randomly sample task parameters $(a, b)$ for modular double exponential $y = b^{a^x} \bmod P$ and generate a sequence of $(x, y)$ pairs without giving explicit information about the task parameters. During the training, the model learns to adapt its prediction $y$ to a query $x$ in-context. We split the train and evaluation set by task parameter pair $(a, b)$, where $a$ and $b$ are sampled from the primitive roots of $P$. We use 80% of all possible combinations of $(a, b)$ during the training, which include all possible individual $a$ and $b$, but not all pairs. The unseen 20% of unseen $(a, b)$ pairs is used as the test set. Throughout the main experiments, we focus on $P = 59$, and we extend our findings to other $P$ values in appendix Appendix B.4.

**In-Context Curricula Design:  Controlling In-context Task Correlations.**    An in-context curriculum is designed by showing subtask examples of the compositional task $b^{a^x}$. As shown in Figure 1a, we show examples of each single exponential task of $a^x$ and $b^x$ followed by the compositional task $b^{a^x}$. Importantly, all $x$ for each task sequence in context are unique, and we do not use the same sequence of $x$ for each subtask and the compositional task. We vary the design of curricula by controlling the length of subtasks and compositional tasks, given the fixed total context length. Using length $m$ for each single exponential subtask and $n$ for the compositional double exponential task yields a total context length $2m + n$, and we will denote $m$-$m$-$n$ to describe each curriculum. We fix the total context length to 48, corresponding to 24 pairs of (x,y). We vary $n$ in range of $[2, 16]$ and corresponding $m$. We train a transformer on these sequences using a next token prediction task for every (x,y) pair in the sequence, rather than only on the final query at the last pair.

While varying the curriculum design, we maintain the importance of the compositional task equal to each single exponential task by controlling the weighting factor for the loss contribution scaled to the length (namely, making it such that the loss from the compositional task is 1/3 of the total loss). Similarly, for a fair comparison of the seen information during the training, in the vanilla setting (0-0-24), the network also sees sequences of a single exponential task as well, with a ratio of 2 to 1 to match the overall weight of the compositional task to other curriculum settings. By doing this, we effectively make it so that the same information is seen in different curricula settings, with the same loss weighting for subtasks and composite task.

The key difference is ***in-context correlations***: as the curriculum length $m$ increases, the in-context correlation between the subtask and compositional task becomes more prevalent (possibly leading to utilization of subtask information), while shorter curriculum length pushes towards learning of the individual function in-context.

## 2.2 MODEL AND TRAINING

We train 8-layer transformers with sinusoidal positional embeddings with a time constant of 120, a hidden dimension size of 128, and 8 heads, using the Adam optimizer, a learning rate of $7.5 \times 10^{-4}$, and a batch size of 512. We trained the models on 10 seeds and trained with $2 \times 10^8$ sequences. See Appendix B.1 for the example loss curve and performance evolution.

## 3 RESULTS

### 3.1 IN-CONTEXT CURRICULA ENABLES IN-CONTEXT GENERALIZATION OF COMPOSITIONAL TASKS

**Zero-Shot Inference on Compositional Task with In-Context Curricula.** We examine the zero-shot inference ability of the model trained with a subtask curriculum. High-zero shot performance on the compositional task, followed by the subtask examples, implies that the model successfully learned to extract the subtask information and utilize it without many-shot examples of the compositional task. We sample 2K sequences from the test parameter combinations $(a, b)$ averaged over 10 seeds and report the zero-shot accuracy in Figure 2 and confirm that the model makes near-perfect zero-shot predictions on the compositional task, compared to the baseline zero-shot performance of the vanilla model, which is essentially random chance level performance since there is no information available in-context yet.

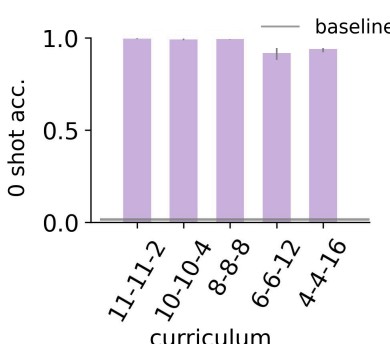

Figure 2: **Zero-shot performance on the compositional task in curricula-trained models.** The model trained with in-context subtask curricula can generalize zero-shot on unseen sequences, suggesting that the model learned to utilize the subtask information from the curricula. Baseline(gray): the zero-shot performance of the vanilla model, which is at the chance level since there is no information yet.

**Different Behavior in Systematic Failure Mode.** We ask if in-context curricula make the model more robust to unseen test sequences, given the same context length. In Figure 3, we compare the accuracy on $n$ compositional task exemplars in each curriculum setting and the corresponding $n$ last exemplars in the vanilla setting across 2K test sequences of $a = 30$ and $a \neq 30$ cases (see Figure 1b and Appendix B.2 for the result throughout the entire context length). The modular nature of the task predicts that $a \neq 30$ test cases are easy to generalize with the few-shot examples, but $a = 30$ becomes challenging. In this case, the information of the subtasks $(a, b)$ from preceding in-context curricula will help generalization by composition. We elaborate on the modular properties in Appendix A. As expected, we found that all models generalize well on $a \neq 30$ test sequences, whereas $a = 30$ shows differential generalization ability. In $a = 30$ cases, we observe higher accuracy across all the curriculum settings compared to the vanilla baseline.

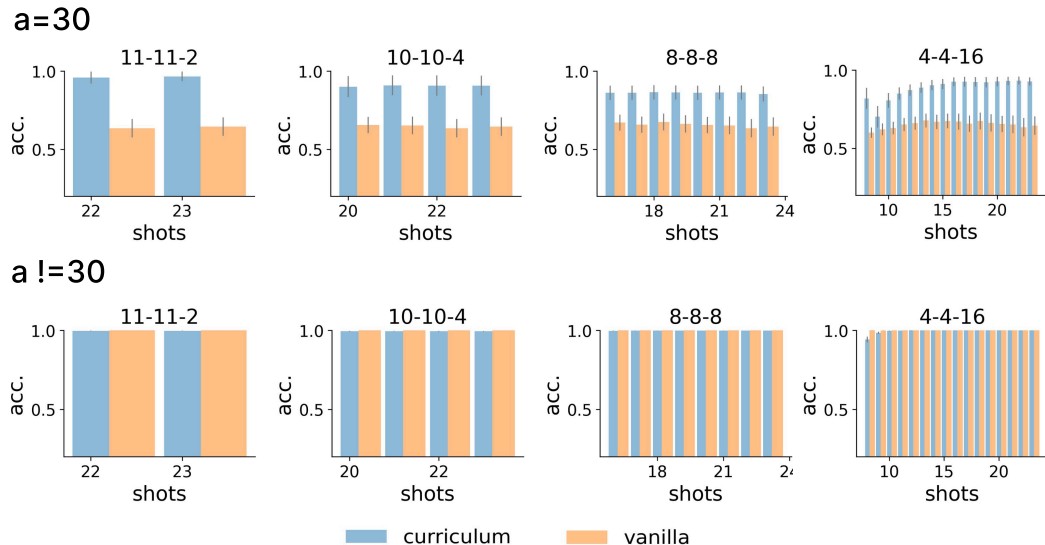

Figure 3: **In-context curricula increase robustness in the challenging test compositional tasks** $a = 30$. In-context error counts of *compositional task* in vanilla model vs. curriculum models in unseen test sequences of $a = 30$ (top) and $a \neq 30$ (bottom). The modular properties predict $a = 30$ to be challenging, while $a \neq 30$ cases are trivial generalizations after a few shots of examples. The leftmost point is zero-shot of the compositional task for the curriculum model. We compare the error counts on the compositional task of curriculum ($m$-$m$-$n$) models to those of the corresponding $n$ last examples from the vanilla model. The curriculum model shows more robust generalization than the vanilla model in challenging $a = 30$ cases, while the vanilla model's accuracy saturates early on in the context. See Appendix B.2 for accuracy over the entire context length and about the $a = 30$ test behavior.

Note that while other curriculum settings exhibit rather consistent errors throughout the compositional task, the 4-4-16 curriculum (rightmost) shows a further increase in errors with more compositional exemplars. This suggests that the model utilizes in-context compositional examples rather than relying solely on the subtask curriculum. Furthermore, a slight performance difference among curricula indicates that varying curricula length results in different model behaviors. We dive deeper into this aspect in Section 3.3- 3.4.

## 3.2 IN-CONTEXT CURRICULA PROMOTE REPRESENTATION OF COMPOSITIONAL SUBTASKS

Based on the above behavioral evidence (zero-shot inference ability and different generalization behavior on challenging test cases), we ask what mechanism is behind this. We hypothesize that the in-context curriculum of the single exponential tasks builds an internal representation of the subtasks, thereby facilitating the model's learning of a subtask composition strategy to solve the double exponential task.

**Linear Probing of Intermediate Computation Values.** To test this hypothesis, we investigated how the model represents intermediate values from constituent subtasks. We trained a linear classifier[1] with the hidden representation of each layer from evaluation sequences to decode the intermediate values required for compositional computation ($a^x \bmod (P-1)$ and task parameter $b$)[2] in compositional task block. We used $80/20$ split of the 1K unseen sequences for the linear probe training and testing of the decoding accuracy. See Appendix C for a control experiment with shuffled labels to confirm the baseline probe performance and more results in other curriculum designs and experimental details.

---

[1]While simple, linear probing is the typical first pass at studying internal representations Gurnee et al. (2023); Nanda et al. (2023b).

[2]Note $b^{a^x} \bmod P = b^{(a^x \bmod (P-1))} \bmod P$.

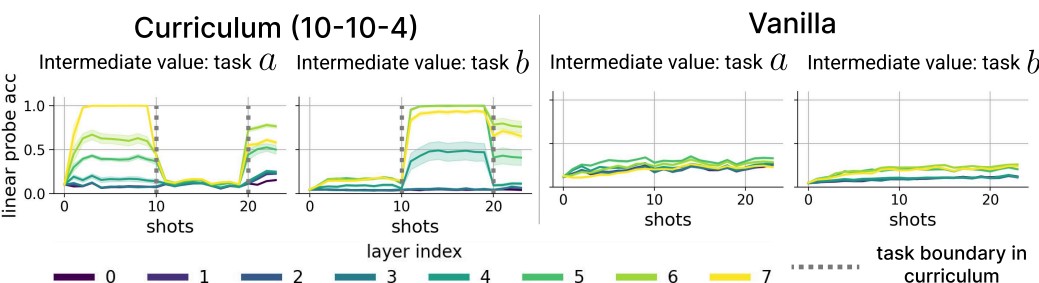

Figure 4: **Linear probe shows difference in intermediate values computation in curriculum and vanilla models.** Linear probe decoding accuracy of intermediate values from the subtasks in the (10-10-4) curriculum (left) and the vanilla setting (right). In the curriculum setting, the intermediate values from the subtasks $(a, b)$ required for compositional computation show high decodability in the corresponding single task block and, importantly, *in the compositional task block (shots 20-23)*, but not in the vanilla model. Noticeably, in the curriculum model, the highest decodability of intermediate values in the compositional task block comes from not the final layer but earlier layers (light green), indicating layer-wise processing of intermediate computations.

In Figure 4, we find noticeable differences in the decoding of intermediate computation values. With in-context curricula, each subtask parameter is well decoded in the corresponding subtask block. Followed by that, the intermediate computation values involving the subtasks $(a, b)$ are highly decodable in the compositional task block (shots 20-23), suggesting that the subtask representation inferred from the curriculum is utilized in the compositional task. Especially, the high decodability at the zero-shot of the compositional task (shot 20) indicates that the model readily utilizes the intermediate computation values inferred from the subtask curriculum to solve the compositional task at zero-shot, aligned with the high zero-shot performance in Figure 2. In contrast, the vanilla-trained model shows lower decoding accuracy of intermediate values required for compositional computation, indicating that training with in-context curricula incentivizes the more compositional representations of the subtasks (at least, in terms of linear decodability).

**Layer-Wise Processing of Intermediate Computations in Curriculum.** Next, we take a closer look at the layer-wise decoding accuracy. In the curriculum model, we observe that the highest decoding of the intermediate computation values in the compositional task is not achieved in the last layer but in the earlier layers (layer 5-6). This suggests the layer-wise processing of subtask information in the compositional block. That is, the representation from the subtask curriculum blocks in the context is transferred and processed in the earlier layers, and then undergoes further computation to combine these representations and perform the compositional task. Additionally, we visualize the attention pattern of the heads in the earlier layers, where we can find heads that attend to the earlier curriculum block from the compositional task block in Appendix D. Collectively, these results indicate that the curriculum-trained model encodes and utilizes the intermediate values required for the compositional task in-context.

**Mismatch Experiment: Causal Behavioral Intervention.** As a causal behavioral intervention, we test the curriculum model with mismatch sequences – single exponential tasks with task parameter $(a, b)$ followed by a double exponential task with mismatching $(a', b')$. If the model is using a subtask composition strategy, we expect the model to fail on this task, as the subtask representations provide incorrect information. In Figure 5, we show that this prediction is consistent in the curriculum model trained with short compositional task block $n$ (11-11-2, 10-10-4, 8-8-8).

With linear probing and behavioral intervention, we show the evidence that the model trained with an in-context curriculum encodes the subtask information in its internal representation and is capable of using it for the compositional task. In contrast, the vanilla model does not necessarily represent such intermediate computation from the subtasks. The mismatch experiment further confirms that having incorrect subtask information causes the curriculum-trained model to fail on the compositional task.

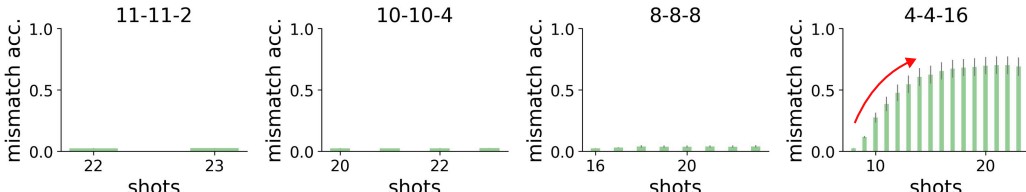

Figure 5: **Mismatch experiment shows causal behavioral evidence of subtask utilization in curriculum-trained models, but it is nuanced with curricula design.** We show model accuracy on the compositional task with $(a', b')$ after giving the mismatching subtasks $(a, b)$. The failure indicates that the model relies on a composition strategy utilizing the subtask information given in the curriculum and thus cannot solve the problem without the correct subtask information. However, with the curriculum with long $n$, such as (4-4-16), the accuracy still increases with few-shot examples of the compositional task after the wrong subtasks, indicating that the model learned the vanilla few-shot learning as well (discussed more in Section 3.3).

### 3.3 IN-CONTEXT CURRICULUM DESIGNS CHANGE THE MODEL'S STRATEGY ON A COMPOSITIONAL TASK

**Mismatch Experiment: Behavioral Signature of Mixed Strategy.** In Figure 5, in the curriculum with long compositional task length (4-4-16), we see performance improvement even when the subtasks mismatch the compositional task. This is not only due to the more examples since the (10-10-4) or (8-8-8) completely fails given 4-8 examples, but the (4-4-16) model shows improvement already from a one-shot example. This implies that the model still makes some inference without the correct subtask information. In other words, the model can learn the compositional task independently, even when the curriculum sequence is provided, without relying on the subtask information,: suggesting it possibly learns both strategies: the subtask composition strategy and the vanilla few-shot learning strategy.

**Linear Probe Detects Mixed and Graded Nature of Strategies.** The sign of mixed strategies becomes clearer in the linear probes. In Figure 6, we show linear probing of intermediate computation from varied curriculum designs. In the curriculum design with long compositional task blocks (6-6-12, 4-4-16), we observe high decodability of intermediate computation at the zero-shot (dotted lines) of the compositional task, but low decodability similar to the vanilla setting in the rest. This implies that the model uses a subtask composition strategy on the zero-shot of the compositional task since the vanilla strategy cannot provide any meaningful inference on the zero-shot, while the model utilizes the vanilla few-shot learning strategy in the rest of the context, as more examples are given. On the other hand, the curricula with shorter compositional tasks (e.g., 11-11-2) consistently show high decodability of intermediate computations across the entire block, indicating that the model predominantly relies on a subtask composition strategy throughout the entire context.

It is notable that the intermediate value decodability is not all-or-none for (8-8-8) or (10-10-4). Rather, it increases continuously as the compositional task length decreases (see green and yellow lines in Figure 6), reflecting that the mark of subtask composition strategy is graded. That is, the possible choice of strategy that a model employs is not merely a binary choice of either compositional or non-compositional, but rather lies on a continuous spectrum.

### 3.4 IN-CONTEXT CURRICULUM DESIGN MODULATES DEVELOPMENT OF STRATEGIES

**Curricula Designs Change in which the Tasks are Learned.** Next, we look into the loss evolution and linear probe across the training phase to understand how the different strategy choices develop. In Figure 7a, we observe a clear difference in the order in which each task is learned when the curriculum design changes. With a long compositional task block (4-4-16), the model is capable of vanilla few-shot learning on the compositional task before the subtasks are learned (pink before gray). Only at the zero-shot, the compositional task loss decreases substantially after the subtask learning (blue after gray). On the other hand, with shorter compositional task length (10-10-4), the rapid learning of the compositional task happens only after the subtask learning (pink and blue after gray). Furthermore, both the zero-shot and the last-shot loss decrease almost simultane-

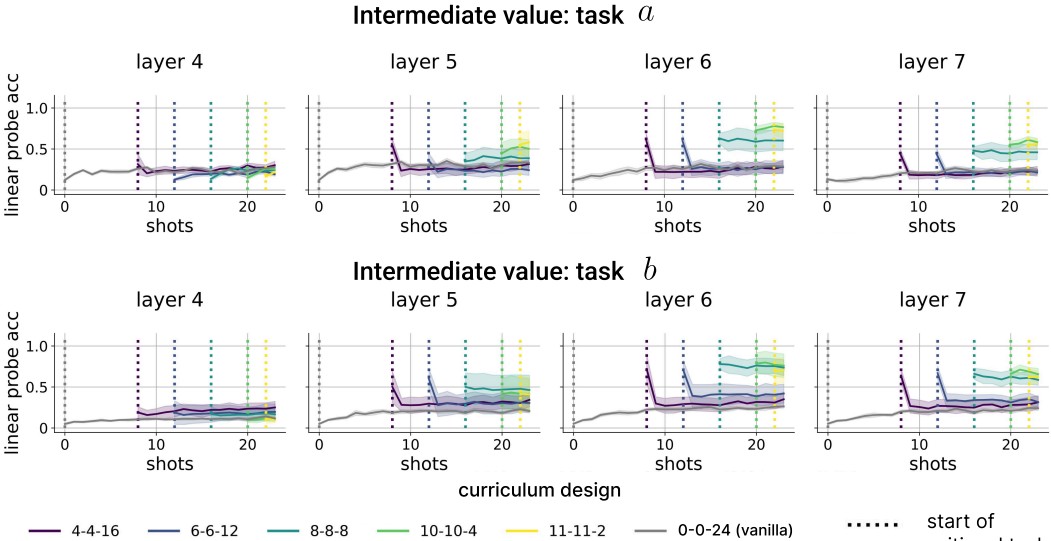

Figure 6: **A complex interplay between the different curricula design and model strategies.** We show the decodability of intermediate computation involving each subtask $(a, b)$ in the compositional task block across different curriculum designs in layers 4-6. At the zero-shot of the compositional task (dotted lines), both intermediate values are highly decodable in longer compositional task curriculum design (6-6-12, 4-4-16), and they decrease afterwards, indicating the existence of both vanilla and compositional strategies. For the shorter compositional task design (10-10-4, 8-8-8), we observe high decodability consistently in the entire block, but the level of decodability is relatively higher with shorter compositional task length (yellow vs. green).

ously, suggesting the subtask information is utilized in the entire compositional task block. It shows that different curricula can change the training dynamics, that is, in which the order of the tasks is learned.

**Linear Probing Strategy Evolution.** We find that the development of the subtask representation in different curricula aligns with the above observation. In Figure 7b, we show a linear probe of the model checkpoints before and after the subtask learning (marked with the colored bars in Figure 7a). In the curriculum (4-4-16), the model can readily solve compositional tasks before learning the subtasks with sufficient examples in-context, and its linear probe shows low decodability of intermediate subtask computation (suggesting a vanilla few-shot learning strategy). The model becomes capable of zero-shot inference shortly after the subtask learning, facilitated by the subtask representation (increased decoding accuracy of intermediate values at the zero-shot). On the other hand, we see that in the curriculum (10-10-4), the compositional task performance rapidly increases only after the subtask learning, and entire compositional task block encodes the intermediate values from the constituent subtasks.

Collectively, these suggest varying the correlational structure between subtask and compositional task given in-context by controlling curriculum design influences, which task to be learned first, and influencing the strategy that the model employs, in this case, compositional computation in-context.

## 4  RELATED WORKS

**In-context learning** has brought significant interest recently, particularly due to the emergent capabilities of LLMs (Wei et al., 2022). The in-context few-shot learning ability of language models (Brown et al., 2020) can be seen as an instance of meta-learned few-shot learning, where the model adapts and generalizes to unseen input examples without requiring gradient updates or explicit meta-training—in contrast to earlier works on meta-learned few-shot learning (Santoro et al., 2016; Vinyals et al., 2016; Wang et al., 2016). Many studies (Chan et al., 2022; Xie et al., 2022; Raventós et al., 2024) have highlighted the importance of data properties for in-context learning. A

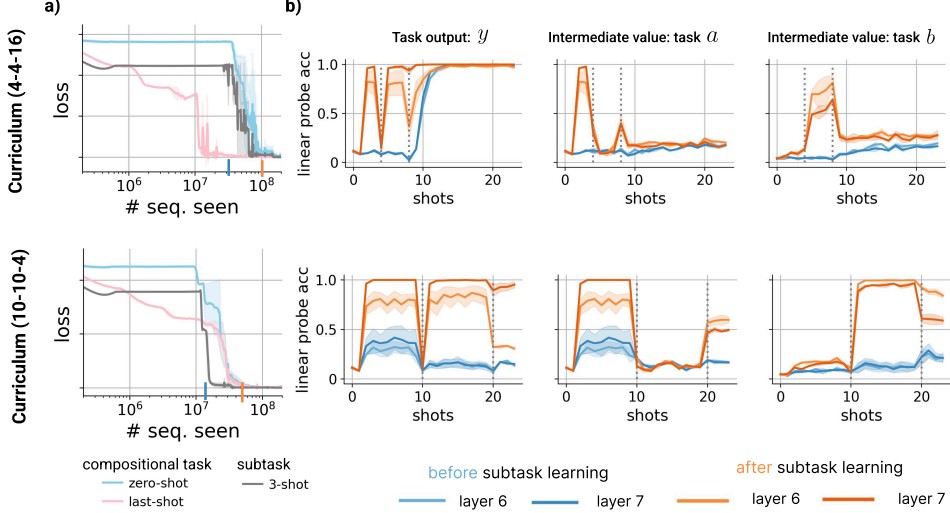

Figure 7: **Curricula design changes task learning orders, which explains the mixed strategy. a)** *Top*: The curriculum (4-4-16) model can solve compositional tasks with vanilla few-shot learning independent of subtask learning (pink before gray). The zero-shot loss decreases sharply only after the subtask learning (blue after gray). *Bottom*: In the curriculum (10-10-4), the compositional task is learned after the subtask learning. The learning of both zero-shot and the last shot happens simultaneously (pink and blue after gray). **b)** Linear probe before and after the subtask learning (blue and orange marks in panel a). *Top*: In the curriculum (4-4-16), we see that the model readily makes good predictions on $y$ of the compositional task after a few-shot examples before learning the subtasks (blue). After the subtasks are learned (orange), the model can infer zero-shot with the subtask representation (increased decodability of $y$ at shots 8-10, and corresponding peaks in decodability of intermediate values). *Bottom*: In the curriculum (10-10-4), the model can solve the compositional task only after the subtasks are learned. See the low decodability of $y$ before subtask learning (blue) and the high decodability of $y$ and intermediate values after subtask learning (orange). See Appendix E.3 for results on other layers.

few studies (Hendel et al., 2023; Todd et al., 2023) have explored how different in-context tasks can be represented in LLMs in the form of task vectors. Russin et al. (2024) show that ICL can match human patterns of compositional and non-compositional behavior in multi-output categorization tasks.

**Compositionality in neural networks** has been a central controversy. Various works have studied how meta-learning can enable systematic compositional generalization in neural networks (McCoy et al., 2020; Lake & Baroni, 2023). Correspondingly, evidence of compositional representation and computation in language models has been studied extensively in scopes ranging from representational structure (Tenney et al., 2019; Soulos et al., 2020), geometric manifolds (Lee et al., 2025) and circuit-level mechanistic interpretability (Geva et al., 2023; Yang et al., 2024; Merullo et al., 2023; Todd et al., 2023).

**Curriculum learning** is critical in learning of humans and animals, well-attested in a body of literature (Skinner, 2019; Elio & Anderson, 1984; Clerkin et al., 2017; Dekker et al., 2022; Lee et al., 2024). While its potential importance has long been acknowledged in machine learning community (Bengio et al., 2009; Wang et al., 2021), the benefit from curriculum has been shown marginal in standard supervised learning benchmarks (Wu et al., 2021).

In this work, we see ICL as a type of learning algorithm and study the effect of curricula in-context learning.

## 5 DISCUSSION

**Importance of Structure in Data.** The types of compositional context structures we have emphasized in this work occur frequently in natural language data; from textbooks to novels, many

documents introduce simpler elements in the beginning that build to yield more complex interactions later. Thus, while many theoretical works on in-context learning focus on presenting IID examples of a single task in context, our work highlights that language models may yield qualitatively different types of in-context learning when the contexts have a curricular compositional structure. Furthermore, our observation of diverse and even mixed strategies emerging from different curricula suggests a rich inner working of in-context learning modulated by different data structures. These findings, therefore, highlight the importance of considering the many types of context structures that may contribute to in-context learning (Lampinen et al., 2024). We hope that our results will encourage more exploration of diverse structures in-context learning, both in controlled settings and at scale.

**Mixtures of Strategies and Spectrum-Like Property of Compositional Generalization.** In Section 3.3, we observe that different curriculum designs lead to the models showing signs of both compositional and non-compositional strategies. Indeed, compositional generalization seems not to be a binary choice but rather a spectrum-like behavior, echoing similar observations in natural language learning (Rabovsky & McClelland, 2020; McClelland, 2015). This suggests a complex interplay between the data structure providing compositional information and the degree of resulting compositional generalization. For example, even when the underlying compositional task structure is the same, depending on precisely how the subtask and compositional task examples are given, different levels of compositional generalization ability can be induced. We also observe that the strategy of the model is linked to the order in which the tasks are learned, which highlights the importance of dynamical aspects of the emergence of in-context learning (Singh et al., 2024b;a; Park et al., 2024; Singh et al., 2025; Yin & Steinhardt, 2025).

**Limitations and Future Directions** Our analysis is limited to controlled behavioral experiments and linear probes. Further analysis with causal manipulation of inner model components (e.g., path patching) would be necessary to gain a more precise understanding of inner mechanism. The present work, based on a toy task, aims to lay the foundations for such explorations by considering a controlled setting in which we can train models from scratch with controlled manipulations of data properties. The natural extension would be to explore the representations of large language models as they learn novel tasks from compositional in-context curricula. Finally, we describe the spectrum of the model strategies in a qualitative manner. More quantitative modeling of the mixed strategies, especially relating them to compositional generalization, would be an exciting next step.

**Ethics and Reproducibility** While our work aims for a fundamental understanding of transformers, we do not anticipate any immediate societal impact from this research. The codebase to reproduce all results will be open-sourced upon acceptance.

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

# Appendix

## A  MODULAR DOUBLE EXPONENTIAL TASK

### A.1  WHY MODULAR DOUBLE EXPONENTIAL TASK?

We study the task $y = b^{a^x} \bmod P$, which decomposes into an inner exponentiation $e(x) = a^x \bmod (P-1)$ and an outer map $y = g^{j\,e(x)}$ once we write $b = g^j$ for a generator $g$ of $(\mathbb{Z}/P\mathbb{Z})^\times$. Note that we limit our $a, b$ to be primitive roots of $P$. A composition of linear/affine operations (e.g., $y = ax + b \bmod P$) is identifiable by two examples of $x - y$ pairs and thus composition does not make the task harder or more complex. On the other hand, the modular double exponential task composes two nonlinear deductions as we show above, namely $e(x) = a^x \bmod (P-1)$ followed by $y = g^{j\,e(x)}$, and it increases the complexity.

### A.2  $a = 30$: CHALLENGING CASE FOR $P = 59$

Now we explain why $a = 30$ is a challenging case for $P = 59$. For any $P$ where $P - 1$ has exactly one odd prime factor ($P - 1 = 58 = 2 \times 29$), the inner exponent $e(x) = a^x \bmod (P-1)$ splits into two periodic group $\mathbb{Z}/2, \mathbb{Z}/29$,

$$e(x) \equiv (\alpha^x, \beta^x), \; \alpha = a \bmod 2, \; \beta = a \bmod 29.$$

The period of the sequence of $y$ is decided by $\beta = a \bmod 29$. For $P = 59$, $a \neq 30$ ends up having a period $T \in 7, 14, 28$ while $a = 30$ is an exceptional case of $T = 1$. That is, for $a \neq 30$, the model can still learn and generalize to the same periodic structure, but for $a = 30$, it is the only case of that specific period, so the model cannot interpolate from other $a$ sequences. We show this behavior extrapolates to $P = 83$, where $a = 42$ is a standout case similar to $a = 30$ for $P = 59$ in Appendix B.4.

## B  ADDITIONAL RESULTS ON MODEL BEHAVIOR

### B.1  EXAMPLE LOSS AND PERFORMANCE CURVES

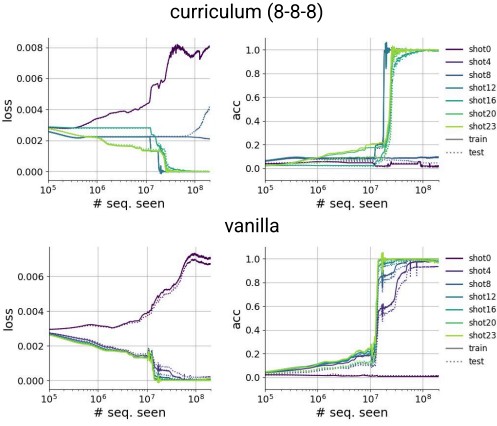

Figure 8: Example loss and performance evolution on the curriculum (8-8-8) model and the vanilla model. In the curriculum model, we observe that the loss at all shots goes down except for shots 0 and 8, which correspond to zero-shot inference of two single tasks, since there is no information available about the task. Furthermore, we observe loss at the later shots in the same task sequence decreases earlier since more information is available in-context. We observe similar for the shot 0 in vanilla model. We also observe multiple plateaus followed by sudden drop of loss in both settings.

## B.2 EXTENDED RESULTS

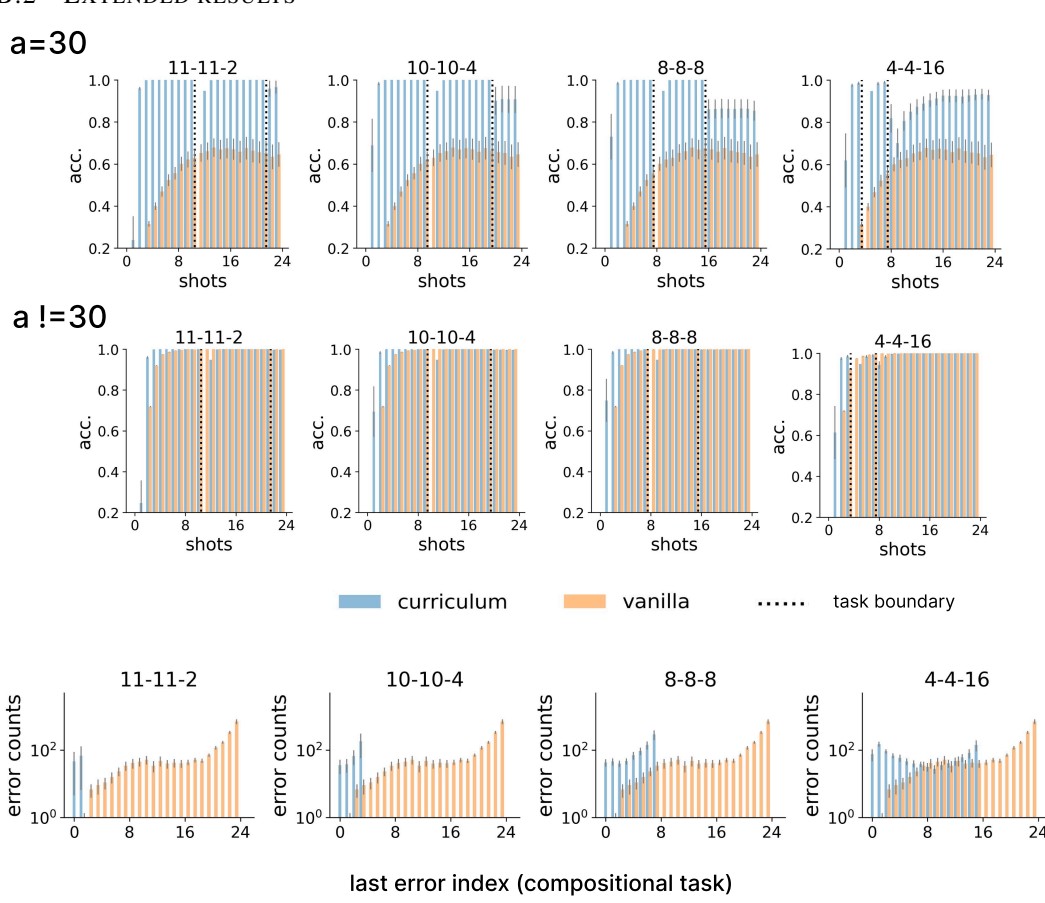

Figure 9: Additional visualization of error counts complementary to Figure 3. **(Top)** Error counts in entire context length across different curriculum designs. **(Bottom)** The counts of the last error occurred at each shot in the compositional task. We observe that the curriculum setting makes fewer errors in total compare to the vanilla models (total counts of each color) reflecting higher robustness of the curriculum models. Furthermore, strong right skewedness of the vanilla model indicates that the model tends to make errors even after many examples suggesting the model is uncertain about the task information.

### B.3 ABLATIONS ON A=30

As explained in the Appendix A, $a = 30$ is the tricky case for generalization due to its own periodicity. While there is already a difference in robustness of the behavior, the periodic nature of the modular double exponential might promote a simple copy operation that is easy for the transformer to learn. In this regard, $x = 0$ is a pre-period step, which is not included in the periodic items, making it another controlled behavioral prediction that makes the curriculum and the vanilla model distinguished. In this ablation, we systematically control the number of $a = 30$ cases (total 28, since $b$ can be one of 28 primitive roots of $P = 59$) in a train/test split over 2 seeds and measure the accuracy at the last compositional task query, and $x = 0$ query in the compositional task.

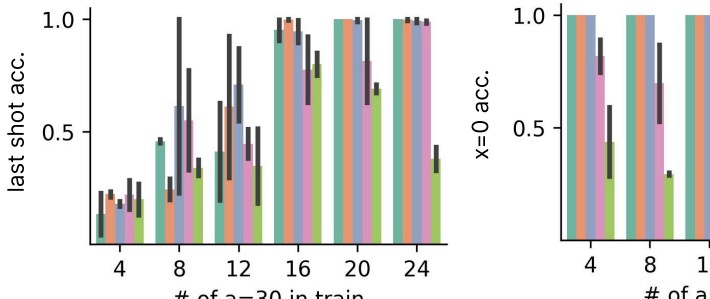

Figure 10: Left: We measure accuracy on unseen $a = 30$ cases with varying number of $a$ in the training set. We find that from 16 cases included in the training set, the model almost fully generalizes to unseen compositional tasks of $a = 30$, while the vanilla model (n=24) shows saturated accuracy with high variance. Right: The failure of the vanilla model becomes clearer in $x = 0$.

### B.4 OTHER P VALUES

We extend our results of robustness with other modulo values, $P = 37, 41$. For generosity, we picked $P = 37, 41$, which does not have one single exceptional period as $P = 59$. For those $P$ values, $x = 0$ and $x = 1$ remain as challenging queries due to ambiguity on $x = 0, 1$ across many $(a, b)$ pairs. We report test accuracy on $x = 0, 1$ cases and the rest. As predicted, $x = 0, 1$ shows differential accuracy (curriculum-trained models keep almost 100% accuracy, while the vanilla model shows saturated accuracy), while the others show all very high test accuracy. The model architecture and the training setup was identical as given in Section 2.

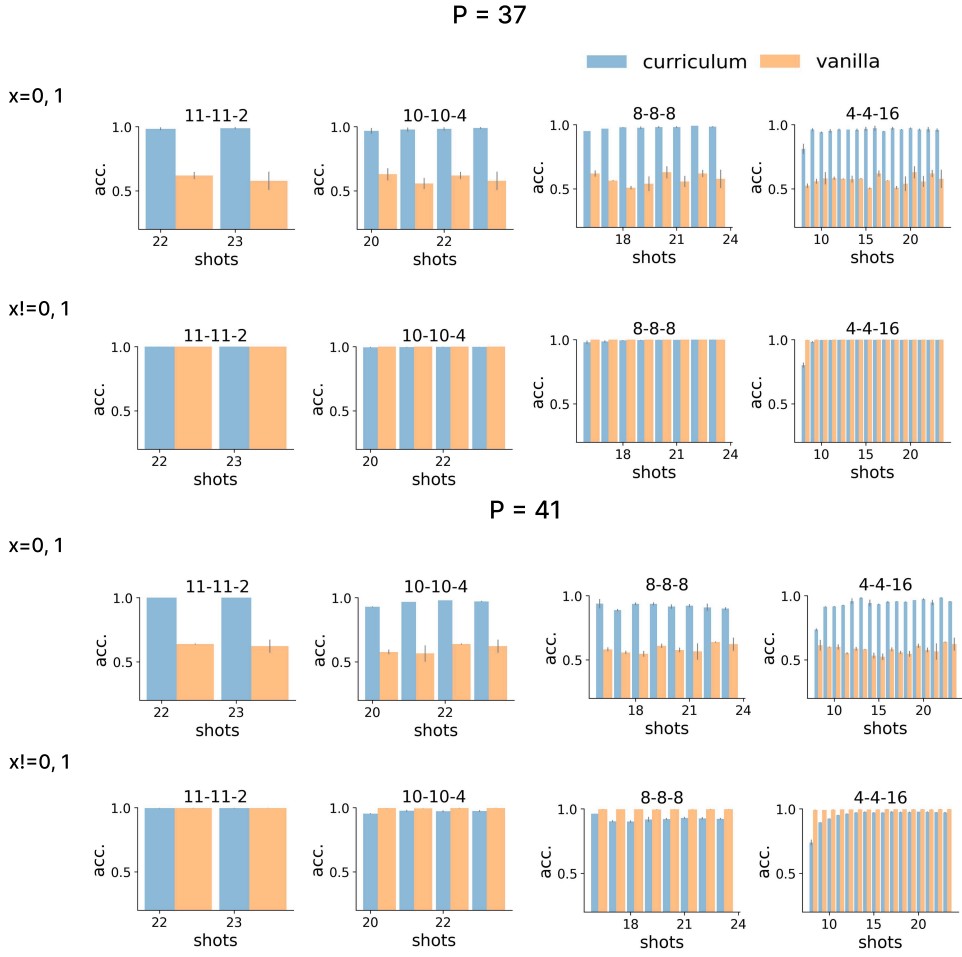

Figure 11: Erros on the compositional task in $P = 37$ and $P = 41$ cases. In both $P$, $x = 0, 1$ becomes ambiguous for the vanilla model and thus causes high errors, while the curriculum model can generalize to even $x = 0, 1$.

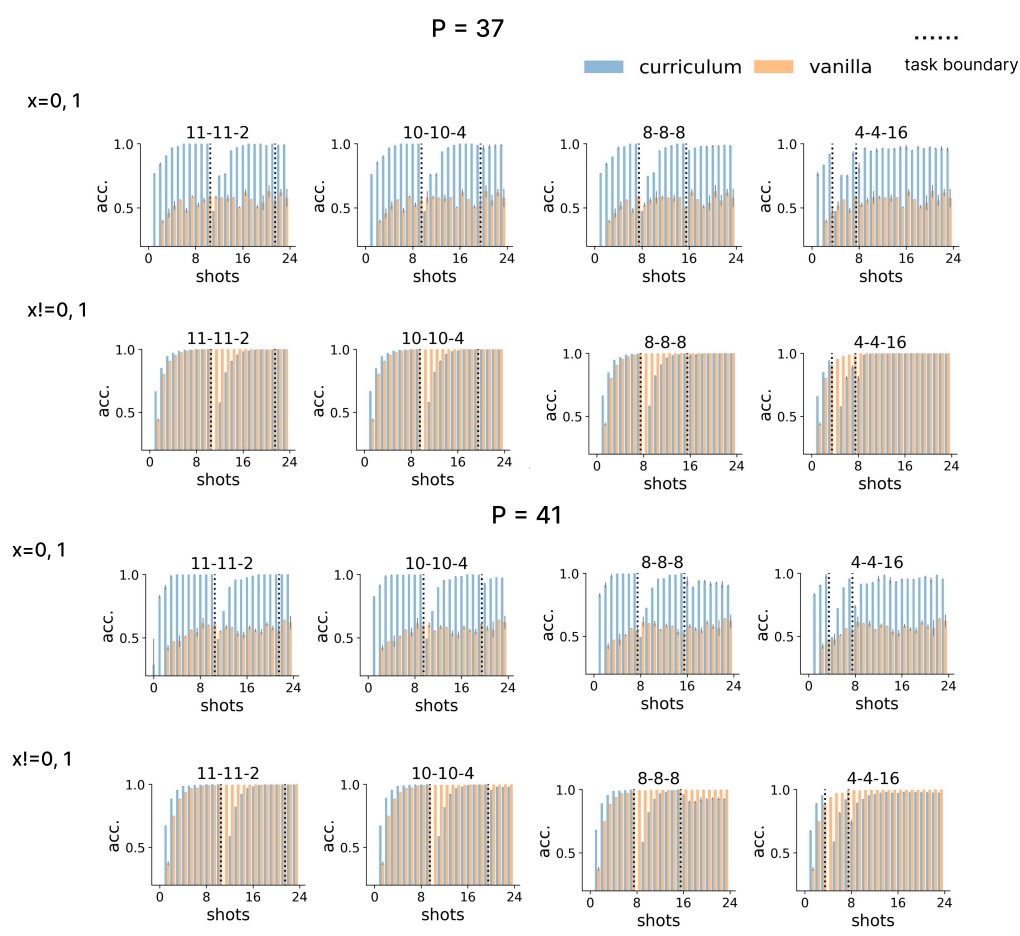

Figure 12: Same as Appendix B.4, but entire context length.

## C    ADDITIONAL RESULTS ON LINEAR PROBE

### C.1    DETAILS OF LINEAR PROBE EXPERIMENTS

We train a linear probe for 1) the corresponding task $y$ at each position, 2) the task computation of $a$, and 3) the task parameter $b$, which are required for the compositional computation of $b^{a^x}$. Since $b^{a^x} \bmod P = b^{a^x \bmod (P-1)} \bmod P$, the intermediate values from task $a, b$ that we try to decode from compositional task blocks are $a^x \bmod P$ and $b$. We train probes for the intermediate values in the subtask block as well. In subtask $a$ block, we simply decode $a^x \bmod P$ (same as target value) and in subtask $b$, we decode $b$. We used unseen 1K test sequences and used $80/20$ split for training and evaluating the linear probe. We used the scikit (Pedregosa et al., 2011) package for the classifier training. The diagram below shows what is decoded in each block.

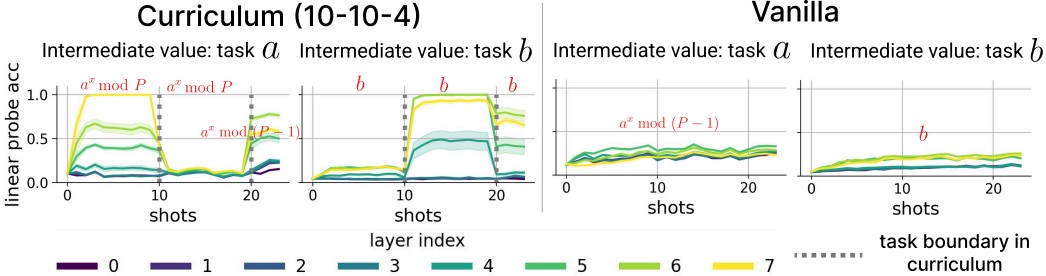

Figure 13: We show what variables we are decoding in each block in main Figure **??**. Since $b^{a^x} \bmod P = b^{(a^x \bmod (P-1))}$, the actual intermediate computation value from task $a$ used for the compositional task is $a^x \bmod (P-1)$, and we can decode this value in the compositional task block when curriculum is given, while less in the vanilla model. Same for $b$.

### C.2    LINEAR PROBE CONTROL BASELINE

We performed a control experiment with shuffled task parameters $(a, b)$ to check the baseline performance and verify that our decoding accuracy is meaningful. The figure below shows that baseline decoding accuracy from shuffled task parameters is almost 0, confirming that our probe decoding accuracy is non-trivial.

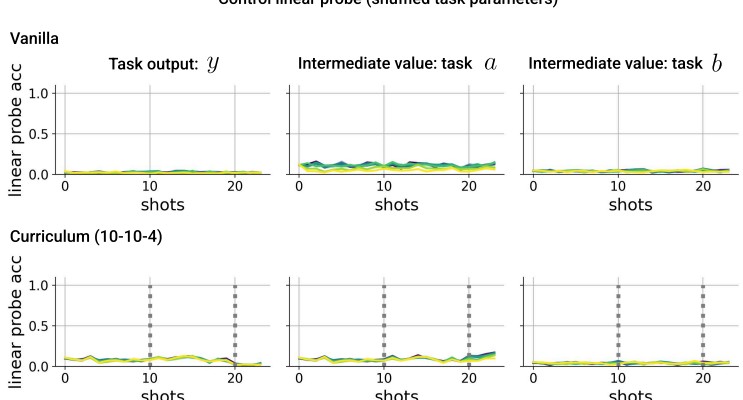

Figure 14: Control linear probe decoding. We used shuffled labels for linear probe training to validate the baseline performance.

## C.3 Linear Probe with Varying Curriculum Designs

Figure 15: Linear probe decoding results for other curricula design. We observe that having an in-context curriculum consistently shows high decodability of intermediate values from the subtasks in the compositional task block. We dive deeper into the specific high decodability at zero-shot in section 3.3- 3.4

# D   ATTENTION PATTERN

We visualize a subset of the attention heads in layers 4-7 in the curriculum-trained model and vanilla-trained model, averaged on 2K evaluation sequences. In the vanilla trained model, the attention pattern is continuous without an outstanding block structure. On the other hand, the curriculum-trained model develops attention heads that show attention patterns from the compositional task block to the curriculum block, which supports our hypothesis that the curriculum subtask representation is utilized in the compositional task.

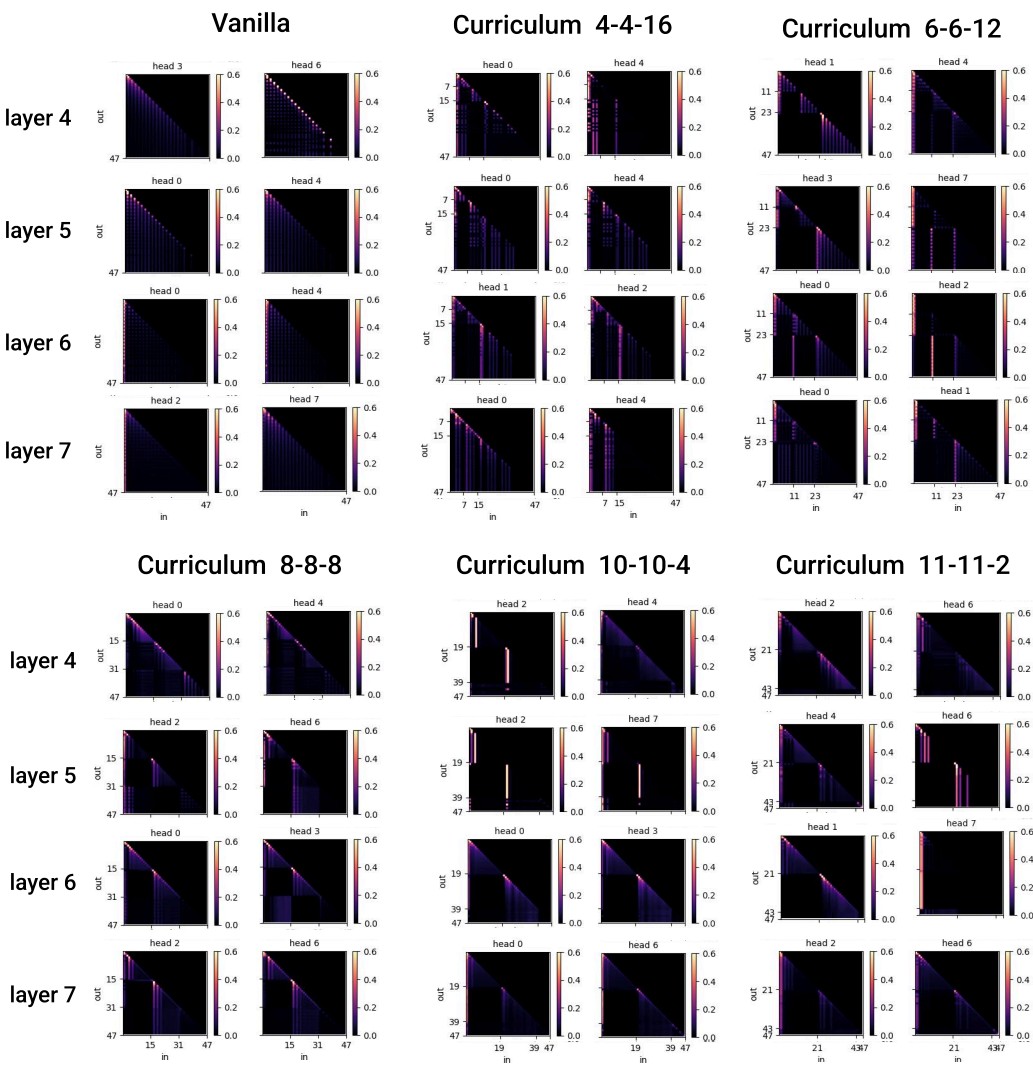

Figure 16: Attention pattern from selected heads in layer 4-7. x-axis shows token position *attending from* and y-axis shows token position *attending to*. We show all 48 token positions of 24 input-label pairs. For curriculum model, each task boundary is marked by ticks on x and y axis. For example, head 0 and head 3 at layer 6 from curriculum (8-8-8) shows a pattern of attending from each subtask to the last compositional task block.

# E ADDITIONAL RESULTS ON MIXED STRATEGIES

## E.1 LINEAR PROBE PER LAYER

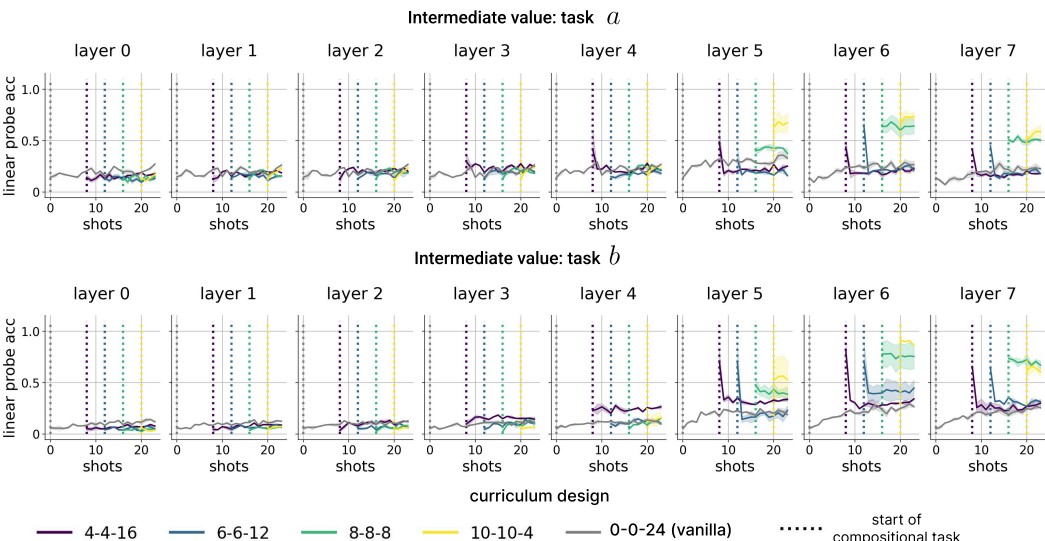

Figure 17: Extended result of main Figure 6. We show the decoding of the intermediate values in the compositional block in all layers.

## E.2 LOSS EVOLUTION

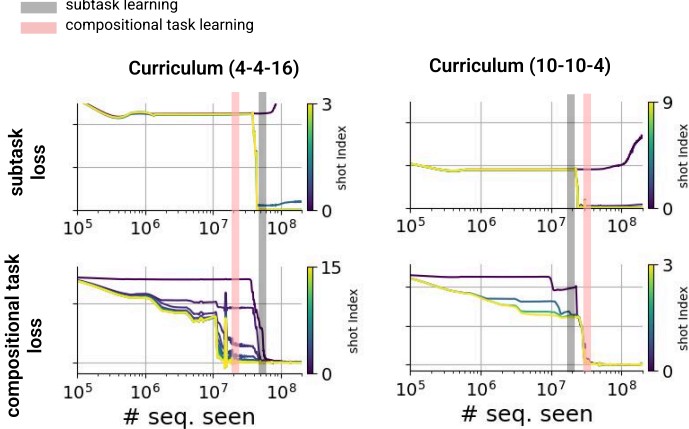

Figure 18: Extended result of main Figure 6. We show loss curve of al shots. The loss curve above and the main Figure 7 are filtered using Savitsky-Golay filter (using scikit implementation Pedregosa et al. (2011)) with length 51 and polynomial order 3.

## E.3   LINEAR PROBE - BEFORE AND AFTER SUBTASK LEARNING

**Curriculum (4-4-16)**

**Curriculum (10-10-4)**

Figure 19: Extended result of main Figure 7.

### E.4 LINEAR PROBE - ADDITIONAL CHECKPOINTS

We visualize the linear probe of curriculum (10-10-4) and (4-4-16) settings at more checkpoints. We first observe that the subtask learning happens much faster in the curriculum. This is because the curriculum (10-10-4) contains more examples of subtasks (single exponentials) compared to (4-4-16).

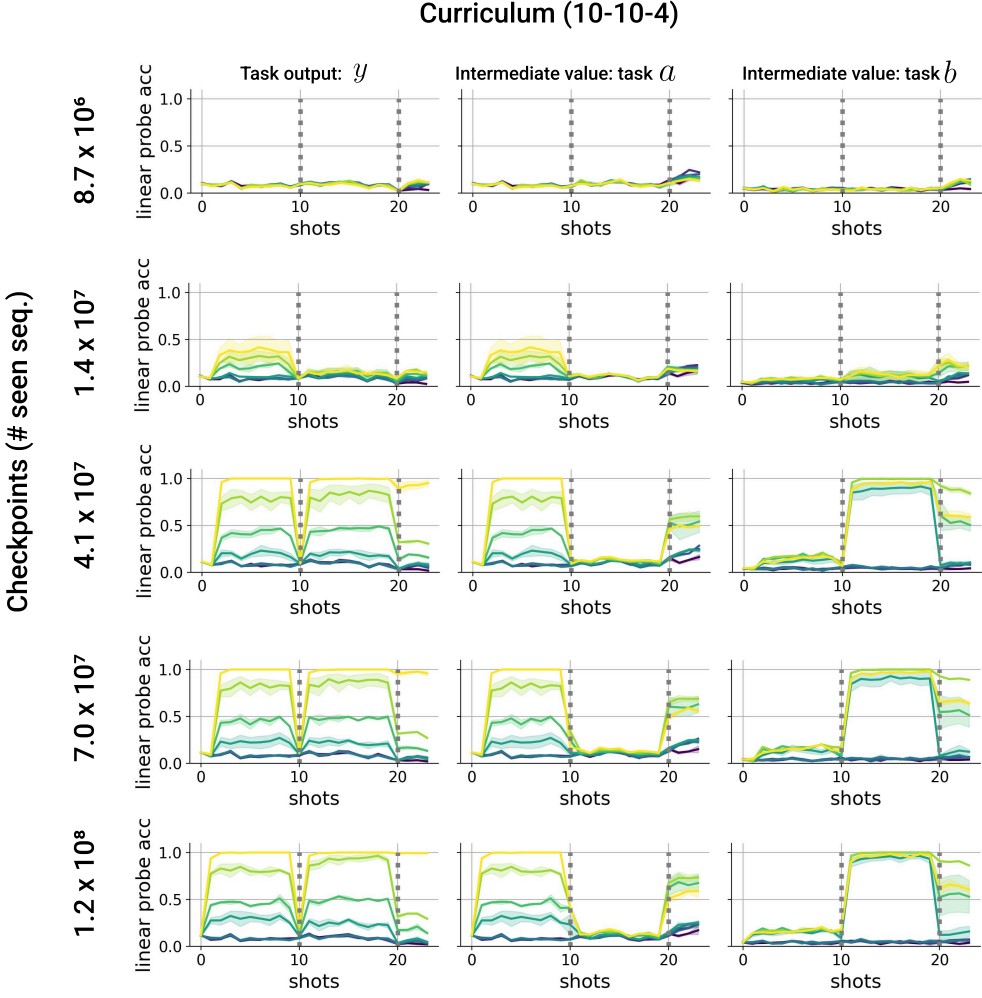

Figure 20: Linear probe of linear probe of curriculum (10-10-4) at more checkpoints.

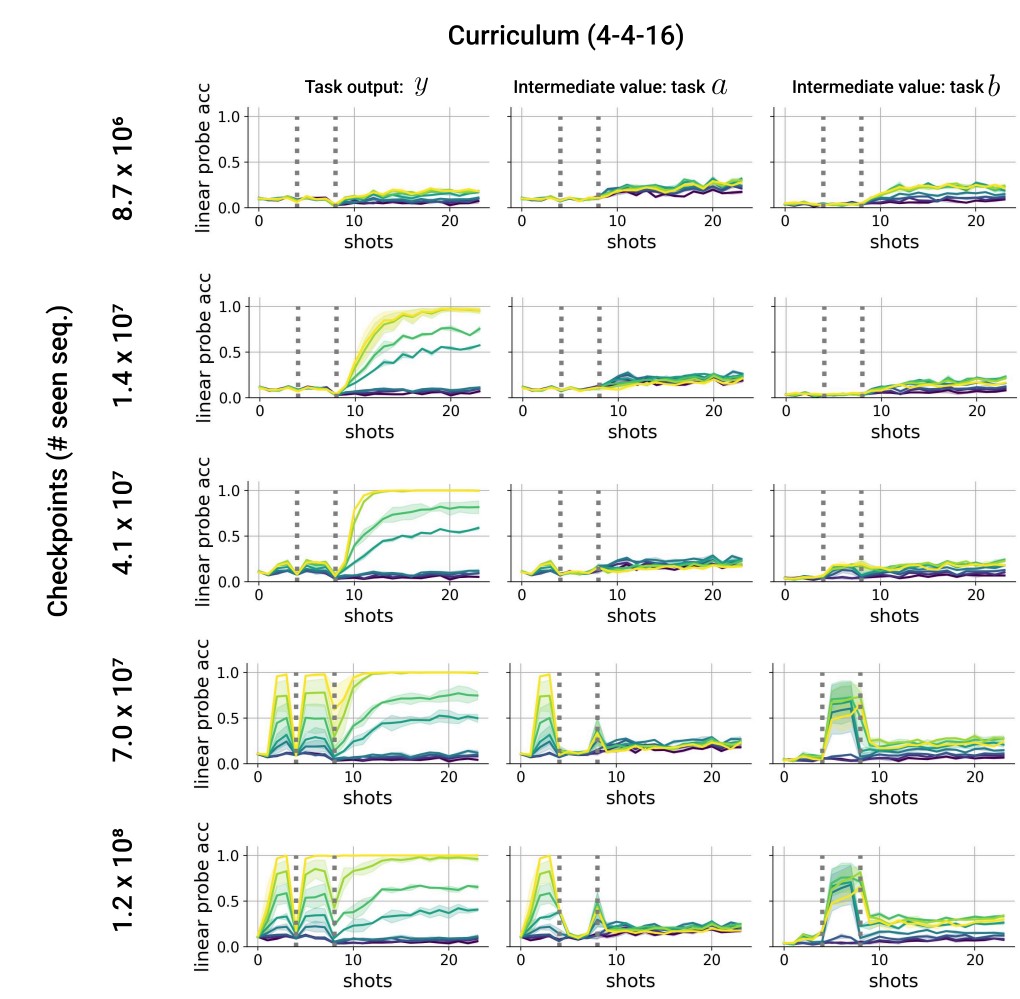

Figure 21: Linear probe of linear probe of curriculum (4-4-16) at more checkpoints.

## F    LLM USAGE

We utilized LLM and AI assistants to refine the manuscript's writing, including grammatical checks and suggestions for phrasing.

