# OpenReview forum: "Distinct Computations Emerge From Compositional Curricula In-Context"
_ICLR.cc/2026/Conference — Submitted to ICLR 2026_

### Official Review · Reviewer_Xz7Y · 2025-10-29

**Soundness:** 3
**Presentation:** 3
**Contribution:** 2
**Rating:** 6
**Confidence:** 4

**Summary:**

The paper studies the compositional ability of small-scale transformers through subtask learning. They provide insights into the mechanism by which the models learn this compositional task and how this is enhanced when given suitable subtasks.

**Strengths:**

(1) The authors choose an interesting and challenging compositional task. To the best of my knowledge, this particular task has not been studied before.
(2) The authors propose algorithms which are well-motivated from empirical observations and then supplement these hypotheses with well-designed experiments.

**Weaknesses:**

(1) The authors should discuss in some detail how their task differs in complexity from other previous works. Eg. the compositional  structure already arises in learning *linear combinations of two variables* $z = a x + b y$ where the model has to learn addition and multiplication. In addition, there should be clear discussions as to the merits and demerits of the choice of this problem.
(2) The authors only perform linear probe experiments. The analysis will be strengthened by causal interference or activation patching experiments.
(3) Due to compute restrictions, it's understandable if the authors could only focus on P < 60. However, the distribution could be skewed for small P, which could lead to inadvertent distributional shifts between train and test sets. One way to mitigate this effect would be to present results for different P, which could give some indication for trends as a function of the modulus. However, due to time constraints, I'd understand if the authors are not able to perform this experiment.

**Questions:**

(1) I suggest putting Related Works section after Introduction.
(2) An interesting direction to take in future, would be to see if the two compositional maps are inverses of each other, is the compositional task ability hampered? Slightly more non-trivial, would be to find functions f(.) and g(.) which give f(g(x)) = ax mod P or some other easy function.
(3) Can the authors describe the exact setting that they used for their zero-shot compositional task experiment in Section 3.1? The high accuracy for all curricula-trained model seems a little surprising.
(4) Do the authors observe any sudden transitions in the accuracies as a function of training time?

---

### Official Review · Reviewer_aLM4 · 2025-10-31

**Soundness:** 3
**Presentation:** 4
**Contribution:** 2
**Rating:** 4
**Confidence:** 3

**Summary:**

This paper studies in-context curricula in a toy task (composition of two exponential functions). Experiments suggest that in-context curricula lead to two strategies: a direct one (that even emerges without presence of the curriculum) and a compositional one that first infers the two individual exponential functions and then the composed task.

**Strengths:**

- I found the paper well written and easy to follow. The figures and captions are nicely done and complement the text well. I think the authors have done a really good job here in terms of the presentation.
- The toy task and experiment design are overall sensible.

**Weaknesses:**

- What is the relevance of in-context curriculum? The standard in-context few-shot learning setup involves no curriculum. Is the curriculum setup a toy model of some specific class of phenomena in real data? The abstract hints at this, but this remains unspecific. Without such a justification, it remains unclear how interesting the results are. This paper [1] seems highly relevant, and I feel it should be cited, but even then I feel a link to one applied paper isn't necessarily enough to justify why this setup is worth studying.

- The paper only reports experiments on toy models trained on a toy task. Neither theory nor experiments on real-world models nor novel analysis techniques are contributed.

- [this is acknowledged by the authors as a limitations, but I nonetheless want to mention this here as a weakness] The presence of distinct computations is only argued via probing and behavioral tests. Even though there by now are standard causal intervention methods for discovering circuits inside transformers (such as path patching), these are not employed in the current paper.

- line 108-110: Can the authors provide some details on how this was implemented? Were the weights adjusted on the fly based on running averages or similar of the losses? Or how was this done? This is important for replicability.


[1] Chen et al, Skills-in-Context: Unlocking Compositionality in Large Language Models, Findings of EMNLP 2024

**Questions:**

- I'm assuming both the models themselves and the linear probes are all trained with categorical output and cross-entropy, not with a continuous output, right?
- In lines 134-135 I'm assuming "zero-shot setup" means that the authors evaluate only on the *first* sample from the compositional task appearing in the prompt, right?
- line 116: "in-context correlations": it's not clear to me what this term means. Is this a formal notion of correlation?
- line 238: "suggesting" -- this is at best very circumstantial evidence. The causal evidence at the bottom of page 5 is more relevant, but I'd advise toning down the claim in line 238.

Minor: line 316: "...Change in which the Tasks..." -- is "Order" missing after "Change"?

---

### Official Review · Reviewer_eBqP · 2025-11-02

**Soundness:** 3
**Presentation:** 2
**Contribution:** 2
**Rating:** 4
**Confidence:** 5

**Summary:**

This paper explores how transformer models develop different computational strategies when trained with compositional in-context curricula. Using a modular double-exponential arithmetic task $ y = b^{a^x} \text{mod}P $, the authors demonstrate that including subtask examples (e.g., $ a^x $, $ b^x $) before composite tasks leads models to form internal representations of intermediate computations and generalize systematically to unseen compositions. Through behavioral evaluation, causal interventions, and linear probing, the study shows that curriculum-trained models exhibit structured, compositional reasoning, unlike vanilla few-shot learners that rely on surface-level correlations. The work provides valuable insights into how *data structure* within in-context learning sequences can qualitatively shape model reasoning and supports the broader hypothesis that compositional curricula promote interpretable, modular internal computations.

**Strengths:**

The formulation of curriculum in-context learning is new and interesting. In practice, when faced with complex reasoning problems, LLMs often break them down into easier subproblems and then aggregate the results at the end. Part of this process can be viewed as curriculum in-context learning, where LLMs must (1) adapt to different subtask problems according to the context and (2) aggregate the subtask conclusions to solve more complex problems. Therefore, I believe the setting considered in the paper is not limited to the in-context learning problem and may lead to a better understanding of LLM reasoning.

**Weaknesses:**

Although the formulation and motivation of this paper are interesting, I believe it would benefit from additional work to strengthen its empirical and theoretical foundations.

1. the experiments are conducted under a fixed modulus $59$ and the same model size, (with limited ablations for $37$ and $41$), which makes it difficult to assess whether the observed phenomena generalize beyond this particular setup. A more systematic exploration across different modular spaces or related compositional domains would strengthen the claims.

2. the analysis relies primarily on linear probing and descriptive visualization; incorporating more direct causal analyses—such as path-patching experiments—could provide stronger evidence that the proposed “subtask composition” representations are indeed functionally utilized.

3. the paper lacks theoretical grounding: it would be valuable to formalize why and when curriculum-trained transformers develop distinct computations, possibly by analyzing architectural biases (e.g., depth, layer specialization) or by contrasting subtask-based and vanilla models under simplified theoretical assumptions.

Overall, the paper has promise but would be substantially stronger with more comprehensive experiments and theoretical insight.

**Questions:**

I feel it hard to understand some parts.

1. It is important to explain why 30 is an important case in the main paper. It is quite confusing at the beginning.

2. Some paraphgraphs are hard to understand: For example, the paragpraph about "In-Context Curricula Design: Controlling In-context Task Correlations." The name "correlation" seems not matching the commonly used "correlation", making it hard to understand.

---

### Official Review · Reviewer_yycS · 2025-11-04

**Soundness:** 3
**Presentation:** 3
**Contribution:** 2
**Rating:** 2
**Confidence:** 4

**Summary:**

This paper studies in-context learning in a compositional task. Specifically, the authors focus on the target function being $y = a^{b^x}$ ; given only $(x,y)$ pairs in-context as well as intermediate values of the form $a^x$, $b^x$. They analyze the performance in both cases with different ratio of examples from each function $a^x,b^x, a^{b^x}$ given in-context. They furthermore analyze the hidden representations of intermediate layers to identify whether the model performs the intermediate computations.

**Strengths:**

1. The paper is clearly written
2. Extensive linear probing analyses that show representational differences — e.g., intermediate values can be identified in mid-layers.
3. Causal mismatch experiments provide evidence that the model relies on subtask information.

**Weaknesses:**

1. Many of the claims made in this paper have already been demonstrated in [1]. In a different but related setting, the authors consider training with 1) $(x,y)$ in-context examples, where $y$ is the output of a ReLU NN, 2) with the intermediate outputs of each layer $(x,y_i)$ also in-context and finally the 3) with also input output pairs of each layer. This setting is intuitively very similar.
[1] also shows better zero-shot performance of 2,3 compared to 1. The authors also perform an analysis of intermediate layers (see Fig 8).
In addition, [2] is related, as the authors demonstrate how models trained on different compositions of functions can successfully compose them using in-context examples.

2. The one shot performance considered in Fig. 2 is not a fair comparison for the baseline, since as I understand it, in the rest of the evaluations the model still sees the partial observations of $a^x$, $b^x$.

3. The experimental setting is synthetic, and the paper does not include any experiments on real datasets.

[1]: Li, Yingcong, et al. "Dissecting chain-of-thought: A study on compositional in-context learning of mlps." arXiv preprint arXiv:2305.18869 (2023).

[2]: Fan, Ying, et al. "Transformers can learn meta-skills for task generalization in in-context learning." NeurIPS 2024 Workshop on Compositional Learning: Perspectives, Methods, and Paths Forward. 2024.

**Questions:**

1. Could the authors compare their work with [1] and [2], and clarify the specific conceptual and technical contributions of this paper relative to those studies?

2. In Fig. 2 and the subsequent evaluations, what exactly is provided as input to each of the models? It would be helpful if the authors could include a few illustrative examples for the baseline, 11-11-2, and, for instance, 6-6-12, under both zero-shot and two-shot settings.

---

### Meta-Review · Area_Chair_BFd2 · 2026-01-06

**Summary:**

This work mostly received negative reviews. Reviewers’ concerns include incremental novelty over prior work, lack of theoretic grounding, synthetic experiments, and limited performance.

**Reviewer Concerns:**

The authors did not submit the rebuttal. Thus, none of the reviewers' concerns are addressed.

**Reviewer Scores:**

Since no rebuttal is submitted, the reviewers did not engage in further discussion. Therefore, the reviewers’ initial scores should be regarded as final. (1) yycS: 2: reject, not good enough, (2) eBqP and aLM4: 4: marginally below the acceptance threshold, and (3) Xz7Y: 6: marginally above the acceptance threshold. Since only one reviewer was positive while the other three were negative, I recommend rejecting the paper.

---

### Decision · Program_Chairs · 2026-01-26

Reject